# Metabolomics in Pulmonary Hypertension—A Useful Tool to Provide Insights into the Dark Side of a Tricky Pathology

**DOI:** 10.3390/ijms241713227

**Published:** 2023-08-25

**Authors:** Pier Paolo Bassareo, Michele D’Alto

**Affiliations:** 1Mater Misercordiae University Hospital, D07 R2WY Dublin, Ireland; 2Children’s Health Ireland at Crumlin, D12 N512 Dublin, Ireland; 3School of Medicine, University College Dublin, D04 V1W8 Dublin, Ireland; 4Pulmonary Hypertension Unit, Dipartimento di Cardiologia, Università della Campania “Luigi Vanvitelli”, Ospedale Monaldi, 80131 Naples, Italy; mic.dalto@tin.it

**Keywords:** metabolomics, metabonomics, pulmonary hypertension, pulmonary arterial hypertension, Eisenmenger syndrome

## Abstract

Pulmonary hypertension (PH) is a multifaceted illness causing clinical manifestations like dyspnea, fatigue, and cyanosis. If left untreated, it often evolves into irreversible pulmonary arterial hypertension (PAH), leading to death. Metabolomics is a laboratory technique capable of providing insights into the metabolic pathways that are responsible for a number of physiologic or pathologic events through the analysis of a biological fluid (such as blood, urine, and sputum) using proton nuclear magnetic resonance spectroscopy or mass spectrometry. A systematic review was finalized according to the PRISMA scheme, with the goal of providing an overview of the research papers released up to now on the application of metabolomics to PH/PAH. So, eighty-five papers were identified, of which twenty-four concerning PH, and sixty-one regarding PAH. We found that, from a metabolic standpoint, the hallmarks of the disease onset and progression are an increase in glycolysis and impaired mitochondrial respiration. Oxidation is exacerbated as well. Specific metabolic fingerprints allow the characterization of some of the specific PH and PAH subtypes. Overall, metabolomics provides insights into the biological processes happening in the body of a subject suffering from PH/PAH. The disarranged metabolic pathways underpinning the disease may be the target of new therapeutic agents. Metabolomics will allow investigators to make a step forward towards personalized medicine.

## 1. Introduction

Pulmonary hypertension (PH) is a multifaceted illness in terms with various etiologies which have in common a significant increase in right ventricular pressure leading to clinical manifestations like dyspnea, fatigue, and cyanosis. If left untreated, PH evolves into irreversible pulmonary arterial hypertension (PAH). In spite of many therapeutic advances, PAH patients are still affected by significant morbidity and mortality [1,2]. The PH classification identifies five different types of the disease and various subtypes [3,4]. For most of the latter, the underlying pathophysiological mechanism is poorly understood from a molecular standpoint, while they are generically characterized by a variable grade of vasoconstriction, endothelial and pulmonary smooth muscle cell hypertrophy, and chronic inflammation resulting in self-perpetuating injury linked to pulmonary vascular disease and unfavorable pulmonary vascular wall remodeling. Conversely, the knowledge of the circulating molecules and metabolites contributing to the different subtypes of PH is limited [5,6].

Regarding the difference between PH and PAH, the first is a generic definition which is used to indicate an increase in blood pressure in the lungs due to diverse causes. The rise in mean pulmonary arterial pressure has to be ≥20 mmHg at rest on right heart catheterization. PH can be suspected on echocardiography as well. Conversely, PAH is a chronic and, so far, incurable illness which causes the walls of the pulmonary arteries to become tight and stiff, thus increasing the blood pressure irreversibly. PAH must be confirmed on right heart catheterization. PH can be detected in at least thirty-seven syndromes which are subdivided into five clinical groups. The wide variety of PH clinical conditions is significantly greater than that of its hemodynamic variety. The latter is classified in pre-capillary and post-capillary PH, which require markedly different therapeutic strategies. PAH is a specific group of severe and uncommon conditions sharing the presence of pre-capillary PH, a similar clinical picture, and almost identical pathological changes in the distal pulmonary arteries [4,7].

System biology, that is, the analysis of complex biological systems by generating large datasets profiling the genomic, proteomic, and metabolic signatures of biological systems across a number of conditions, represents an emerging tool to discover novel pathogenetic mechanisms promoting cardiovascular diseases like PH [8]. The “omic” approach is becoming more and more used. The reality is that multiple “omic” approaches are possible, namely, (1) genomics, i.e., the measurement of DNA sequence modifications; (2) transcriptomics, i.e., the measurement of RNA expression; (3) epigenomics, i.e., the measurement of DNA modifications not entailing sequence modifications, which act on RNA expression; (4) proteomics, i.e., the measurement of protein expression or of protein chemical variations; and (5) metabolomics, i.e., the measurement of metabolites concentrations [9].

Metabolomics is a laboratory technique capable of providing insights into the metabolic pathways that are responsible for a number of physiologic or pathologic events through the analysis of a biological fluid (such as blood, urine, and sputum) using proton nuclear magnetic resonance spectroscopy or mass spectrometry [10]. The first technique provides an estimation of the mass-to-charge ratio of ions in a small sample of a biological fluid. The latter is capable to characterizing low-molecular-weight molecules holding a nuclear magnetic resonance nucleus in a biological fluid. Mass spectrometry analysis is probably faster than that with proton nuclear magnetic resonance spectroscopy and can be used to evaluate tissue samples as well. Nevertheless, its sensitivity is somewhat low [11,12]. Based on these premises, metabolomics may be a useful tool to identify the molecular pathways underpinning the development of the various forms of PH and their progression to irreversible PAH or Eisenmenger syndrome [13]. The emerging role of metabolomics in shedding light on new circulating markers of PH has emerged ten years ago [14]. Using metabolomics seems to be a reliable way to understand and treat pulmonary vascular disease [15]. This review was carried out to provide an overview of the research papers published so far on the application of metabolomics to PH/PAH analysis.

## 2. Materials and Methods

### 2.1. Search Strategy

A literature search was carried out in the electronic data banks of PubMed, Scopus, and Web of Science from their inception up to 2 June 2023. The MeSH (Medical Subject Headings) terms “pulmonary hypertension”, “pulmonary arterial hypertension”, “Eisenmenger”, “metabolomics” and “metabolomics/metabonomics” were searched. All studies focused on metabolomics applied to PH, PAH, and Eisenmenger were examined in this review, without any restriction with respect to the year of publication. Nevertheless, a few exclusion criteria were applied, regarding (a) research papers not discussing the selected topic, (b) reviews and case reports, (c) manuscripts not written in English, (d) duplicated papers, and (e) manuscripts not obtained from libraries for full-text check.

### 2.2. Study Selection

Each author examined the identified abstracts and judged their eligibility. Full texts were assessed each time that all the reviewers of the abstracts thought that they might meet the inclusion criteria.

### 2.3. Data Extraction

Original works on metabolomics applied to the study of all forms of PH/PAH were searched. Overall, 544 manuscripts potentially satisfying the inclusion criteria were selected. Three hundred and thirteen articles (eighty-four of which were duplicated papers) were excluded after checking their title and abstract. Thus, 231 studies were further checked for full-text assessment. A hundred and forty-six articles were excluded as they did not match the inclusion criteria or were not obtained from online libraries. Thereby, 85 studies matched the inclusion criteria, and 11 more manuscripts were added after looking into the references of the first 85. A PRISMA flowchart of the study selection process is displayed in the Appendix A.

## 3. Results

### 3.1. Pulmonary Hypertension

#### 3.1.1. PH Pathogenesis

In a rodent model of PH triggered by hypoxia, significant shifts in amino acids metabolism were noted. In fact, the levels of branched-chain amino acids were lower, whereas those of intracellular amino acids were increased in microvascular endothelial cells. There was incretion instead of the utilization of amino acids through the pulmonary microvasculature, and an oxidized glutathione gradient was detected trough the pulmonary vasculature, thus implying increased glutamine uptake (glutamine is a source of glutathione) [16]. Six blood metabolites (histidine, alanine, serine, asymmetric dimethylarginine, 2-hydroxybutyric acid, cystathionine) were recognized in hypoxia-related PH rodents [17].

The enrollment and polarization of inflammatory monocytes and macrophages in the areas surrounding the pulmonary arteries (adventitia layer) is a feature of PH. Mass spectrometry-based metabolomics analysis showed a rise in free NADH concentrations as well as the over-expression of a metabolic sensor and transcriptional co-repressor, i.e., C-terminal binding protein 1 [18].

Isotope metabolomics with ^13^C-labeled glucose revealed that the pulmonary artery smooth muscle cells displayed raised glucose uptake/utilization by glycolysis and the pentose shunt, but no modifications in glutamine or fatty acid uptake/utilization. In addition, pulmonary artery endothelial cells showed raised proximate glycolysis pathway intermediates, decreased pentose shunt flux, raised anaplerosis from glutamine, and reduced fatty acid beta-oxidation [19]. Key metabolic pathways contributing to pulmonary vascular remodeling and RV dysfunction include fructose catabolism, arginine–nitric oxide metabolism, tricarboxylic acid cycle, and ketones metabolism [20].

#### 3.1.2. Specific PH Subtypes

Preterm babies are often affected by bronchopulmonary dysplasia with associated PH (Group 1) [21]. Quite surprisingly, a umbilical cord blood metabolomic analysis from preterm newborns showed molecules reflecting dyslipidemia, which were associated with the presence and severity of bronchopulmonary dysplasia [22]. Persistent pulmonary hypertension of the newborn can be triggered by gestational long-term hypoxia. In an animal model of the disease, a targeted metabolomic analysis in the pulmonary arteries of fetal sheep revealed an association of the disease with reduced concentrations of omega-3 polyunsaturated fatty acids and related lipid mediators and increased concentrations of some omega-6 metabolites, including 15-keto prostaglandin E2 and linoleoylglycerol [23].

PH induced by drugs and toxins is classified in Group 1 [3]. In monocrotaline-triggered PH, the arginine, pyrimidine, purine, and tryptophan metabolic pathways were altered [24].

Pulmonary vein stenosis is a rare etiology of PH (Group 1). A swine model of the disease was created by pulmonary vein banding to mimic PH. Significant metabolic disturbances were detected in the pulmonary upper lobes, mostly regarding fatty acid metabolism, reactive oxygen species, and the extracellular matrix. Some changes were found in the lower pulmonary lobes, regarding purine metabolism as well [25].

PH triggered by heart failure (Group 2) is characterized by the overexpression of long-chain acylcarnitines, acetylcarnitine, and monounsaturated fatty acids. A statistically significant correlation with N-terminal pro-brain natriuretic peptide, a biomarker capable of predicting the outcome of cardiac insufficiency, was detected as well [26]. A metabolomic study was carried out with the aim of looking for different blood metabolites whose levels are different in PH subjects in comparison with healthy controls and among patients with different forms of PH linked to left heart disease (Group 2). The PH patients in the study were subdivided into three groups: precapillary PH, postcapillary PH, and combined precapillary/postcapillary PH patients. Raised concentrations of long-chain acylcarnitines were found in patients with all PH forms in comparison with healthy controls, but they did not differentiate between patients with combined precapillary/postcapillary PH and those with postcapillary PH. In PH subtypes analysis, combined precapillary/postcapillary subjects had reduced concentrations of molecules linked to urea cycle amino acids and short-chain acylcarnitines in comparison with healthy controls and with patients with postcapillary PH. The latter associations were weak, however. These results imply that different PH forms share a likely unique metabolic pathway [27].

Congenital diaphragmatic hernia is characterized by the leakage of the abdominal viscera into the thorax amid fetal life. This search for space in the thoracic cavity leads to lung hypoplasia and reduced vascular development, which can cause severe PH [28]. The latter is classified in Group 3 PH due to the developmental abnormalities it presents. Alterations in the levels of lactate, glutamate, and adenosine 5′-triphosphate were found. They highlight changes linked to oxidative stress, nucleotide synthesis, amino acid metabolism, glycerophospholipid metabolism, and glucose metabolism [29]. Also chronic obstructive pulmonary disease-associated PH, one of the most common subtypes of the disease, is classified in Group 3. A metabolomic analysis carried out with ^1^H nuclear magnetic resonance spectroscopy on serum and exhaled breath condensate detected unbalanced lactate and pyruvate levels in both fluids, which were linked to the pulmonary artery pressure calculated during echocardiography [30]. Chronic exposure to high altitude can cause PH development. The latter belongs to Group 3 in the PH classification. ^1^H NMR metabolomics was applied to analyze serum in an animal model of the disease. 18β-glycyrrhetinic acid significantly reduced the pulmonary arterial pressure and malondialdehyde concentration and increased the glutathione peroxidase and superoxide dismutase activities, thus exerting an antioxidant and anti-inflammatory effect and restoring several metabolic pathways (like energy, amino acid, and lipid metabolisms) [31]. Macitentan was used to treat a group of rats in which high-altitude-triggered PH was induced. Metabolomics showed that the purine and arginine/proline metabolic pathways were improved by macitentan, which regulates the expression of xanthine oxidase in the purine metabolic pathway and the activity of arginine 1 and arginine 2 in the arginine/proline metabolic pathway [32].

Pulmonary embolism (PE) is a medical emergency that can cause the onset of PH and right ventricular dysfunction [33]. Chronic thromboembolic pulmonary hypertension (CTEPH) is included in Group 4 in the PH classification [3]. With the aim of gathering more information on PE and PH, PE was induced in white pigs. Their blood was analyzed before and following PE induction. Metabolomic fingerprints were acquired from the plasma with liquid as well as gas chromatography-based mass spectrometry. Twenty-eight compounds were identified by liquid chromatography, and nineteen compounds from gas chromatography. Most of these metabolites are correlated with energy imbalance in hypoxic situations, such as glycolysis-derived metabolites, ketone bodies, and tricarboxylic acid cycle intermediates, along with some lipidic mediators which may be involved in cellular signaling, like sphingolipids and lysophospholipids. These results represented a progress in the understanding of the pathophysiological processes caused by PE and triggering CTEPH [34]. The study was repeated in humans. Fasting blood from a peripheral vein was collected. Three hundred and sixty-two metabolites were found to be different between CTEPH patients and healthy individuals: 178 proved to be present at a higher level, and 184 at a lower level. The levels of fatty acids, glycerol, acyl carnitines, beta-hydroxybutyrate, amino sugars, and modified amino acids and nucleosides were increased, whereas those of acylcholines and lysophospholipids were reduced. Overall, these results imply an increase in lipolysis, fatty acid oxidation, and ketogenesis [35].

Sildenafil is a well-known drug which is administered to treat PH. Purine biosynthesis plays a key role in adverse pulmonary vasculature remodeling associated with PH. While sildenafil alone only modestly reverses purine biosynthesis, a combination therapy with histone deacetylase inhibitors showed inhibitory effects on purine synthesis, representing a novel and, potentially, more efficacious approach against the constriction of blood vessels and their adverse remodeling [36].

The metabolic features of the distinct subtypes of PH are summarized in Table 1.

#### 3.1.3. New Pathogenetic Mechanisms

Other kind of diseases are often seen in patients with PH. In this respect, the serum concentrations of arachidonic acid metabolites in humans suffering from hyperthyroidism were studied by means of mass spectrometry. The increased level of thromboxane A2 in hyperthyroid patients might at least in part explain the PH frequently found in hyperthyroid patients [37].

Another study aimed at examining the role of hyperglycemia in PAH by quantifying and detecting the metabolomic alterations in smooth muscle cells in a situation of hyperglycemia. Untargeted metabolomics with liquid chromatography and mass spectrometry was used in a diabetic-like condition represented by an in vitro culture of pulmonary artery smooth muscle cells in a hyperglycemia-like setting. The results demonstrated that the cells in the presence of high glucose levels proliferated more quickly, thus generating high levels of reactive oxygen species. Cells cultured in high glucose displayed modifications of the carbohydrate pathway, mostly, the pentose phosphate pathway, by the oxidation pathway. The levels of amino acids like aspartate and glutathione were diminished in the hyperglycemia-like setting, whilst those of glutathione disulfide, N6-Acetyl-L-lysine, glutamate, and 5-aminopentanoate were increased. The lipids levels, including fatty acids and glycerophospholipids, were adversely affected in the majority of the cases [38].

#### 3.1.4. Right Ventricular Function

By combining a metabolomic analysis with a cardiac magnetic resonance imaging-derived three-dimensional model of right ventricular geometry and contractility, six metabolites proved to be significantly correlated with elevated wall stress, including raised concentrations of tRNA-specific modified nucleosides and fatty acid acylcarnitines and reduced concentrations of sulfated androgen [39].

### 3.2. Pulmonary Arterial Hypertension

#### 3.2.1. PAH Pathogenesis

Although the expression of many genes and proteins has been widely studied in PAH, the mechanisms responsible for its development and progression from PH remain poorly known. Using a combination of liquid and gas chromatography-based mass spectrometry, disarranged glycolysis, increased tricarboxylic acid cycle, as well as fatty acid metabolites with impaired oxidation were detected in the lungs of patients with PH. This led to increased ATP synthesis, necessary for the adverse pulmonary vessels’ remodeling responsible for the degeneration of PH into PAH [40].

In an animal model of PAH associated with right ventricular failure, metabolomics showed that the levels of of oxidized glutathione, xanthine, and uric acid were raised, thus increasing the release of reactive oxygen species by the enzyme xanthine oxidase. An excessive reactive oxygen species formation is known to modify both the structure and the function of pulmonary arteries walls. A 30-time lower concentration of alpha-tocopherol nicotinate, in line with oxidative stress reducing antioxidants, was found as well [41].

A metabolomic study with liquid and gas chromatography along with mass spectrometry detected 16 metabolites (especially, threitol and aminomalonic acid) that appeared as specific markers of the disease. All of them were linked to energy imbalance and included glycolysis-derived molecules, as well as molecules participating in fatty acid, lipid, and amino acid metabolisms [42]. Similar findings were obtained from the metabolomics analysis of exhaled breath. The latter was tested with gas chromatography associated with mass spectrometry [43].

While PAH resulted in a glycolytic shift, metabolomics discovered that the treatment with exosomes resulted in mitochondrial function and oxygen consumption improvement. The therapy with exosomes increased the expression of pyruvate dehydrogenase as well as of glutamate dehydrogenase 1 in pulmonary artery smooth muscle cells, connecting the exosome therapy with the tricarboxylic acid cycle. This may represent a novel treatment of PAH [44].

Specific metabolic pathways, which, taken together, contribute to pulmonary arteries remodeling in PAH, were identified by analyzing the lung tissue of subjects with advanced PAH by means of a combination of liquid and gas-chromatography-based mass spectrometry. A disarray in arginine pathways with raised nitric oxide and reduced arginine, increased sphingosine-1-phosphate, and heme metabolites with modified oxidative pathways were noted in the lungs of patients with severe disease [45].

An extensive research on plasma metabolites with ultraperformance liquid chromatography–mass spectrometry found raised concentrations of transfer RNA-specific modified nucleosides, tricarboxylic acid cycle intermediates (malate, fumarate), glutamate, fatty acid acylcarnitines, tryptophan, and polyamine metabolites and reduced concentrations of steroids, sphingomyelins, and phosphatidylcholines, as hallmarks distinguishing patients suffering from idiopathic PAH and various forms of heritable PAH from heathy controls. Half of these metabolites were related to the disease outcome [46]. Reduced blood levels of small HDL molecules rich in apolipoprotein A-2 content carrying fibrinolytic proteins appeared to be linked to adverse events in subjects with idiopathic and heritable PAH, as testified by another metabolomic study [47]. Explanted lung tissues from patients suffering from PAH were analyzed with a combination of liquid chromatography and mass spectrometry. Statistically significant differences were found when comparing these subjects with healthy counterparts. The higher the thymine level, the better the patient prognosis. A direct proportion was found between the spermine level and the patients’ cardiac function [48].

Again, identifying and targeting many of the potentially involved pathways could be a more useful way of acting than focusing on a single pathway. So, metabolomic analyses on human pulmonary microvascular endothelial cells confirmed the previously reported rise in aerobic glycolysis. In addition, notable upregulation of the pentose phosphate pathway, stimulation of the nucleotide salvage and polyamine biosynthesis pathways, decreased activation of the carnitine and fatty acid oxidation pathways, significant impairment of the tricarboxylic acid cycle, and failure of anaplerosis were detected [49]. In summary, the main metabolomic features of PAH are the overactivation of glycolysis and the reduced activation of mitochondrial respiration, similarly to what seen in cancer [50].

Lung metabolomics revealed that leucine and isoleucine removal, valine, leucine, and isoleucine synthesis, tryptophan metabolism, and aminoacyl-tRNA synthesis play a role in the development of PAH, with hydroxyphenyllactic, isopalmitic acid, and cytosine being crucial factors [51].

In a mouse model of PAH, a metabolomic analysis of lung specimens identified a significant impairment in the glycolytic pathways, with the overexpression of glutamine metabolism and disarrangements in lipid metabolism. In addition, an imbalance in glycine and choline metabolism was found in lung samples. Metabolic reprogramming was also detected in right ventricular specimens with raised levels of lactate and alanine, which are at the end products of glycolytic oxidation. Glutamine and taurine proved to be related to specific ventricular hypertrophy characteristics on imaging [52].

Reduced levels of protective molecules (such as butyrate and propionate) and increased levels of pathogenic metabolites (like proinflammatory mediators) were found in the gut metabolome of rats suffering from left pulmonary artery ligation-induced PH. Furthermore, the modified metabolome was significantly related to a disarranged microbiome in the gut as well as in the lungs [53].

Distinct plasma metabolomic molecules were linked to different outcomes. Specifically, polyamine and histidine were related to right ventricular dilation, mortality rate, N-terminal pro-B-type natriuretic peptide, and 6 min walk distance. Acylcarnitine was related to N-terminal pro-B-type natriuretic peptide, and 6 min walk distance. Sphingomyelin was related to RV dilation and N-terminal pro-B-type natriuretic peptide [54].

Metabolomics is a promising approach to shed light on potential anti-PAH drugs. Osthole, a natural compound, is one of these medications. It has the capacity to control PAH in rodents, but its molecular mechanism is still poorly understood. Liquid chromatography–mass spectrometry showed that osthole exerts his beneficial action by inhibiting the expression of Sphk1/S1P through the downregulation of the microRNA-21-PI3K/Akt/mTOR signaling pathway [55].

Kaempferol, a natural flavonol, seems to be able to slow down the onset of PAH. Its beneficial effect on pulmonary arteries remodeling requires the upregulation of the amino acid and arachidonic acid metabolism, as revealed by metabolomics [56].

Metabolomics profiling demonstrated that inhibition of lysine kinase 1 improved the right-ventricle metabolism and, in turn, its systolic function [57].

Treatment with luteolin, a dietary supplement, returned the altered arachidonic acid, amino acids, and tricarboxylic acid cycle levels close to normal. Luteolin partially reversed the adverse pulmonary vascular remodeling in PAH rats by blocking the excessive and rapid increase in the number of pulmonary artery smooth muscle cells [58].

CTEPH is often treated by pulmonary endarterectomy (PEA). The plasma metabolome was investigated before and after PEA. In responders to PEA, changes in tryptophan, sphingomyelin, methionine, and Krebs cycle metabolites were noted [59].

#### 3.2.2. Specific PAH Subtypes

An increased number and resilience to apoptosis of pulmonary arterial vascular smooth muscle cells are pivotal in the irreversible pulmonary vascular remodeling responsible for PAH development. Pulmonary artery hyperplasia results from pulmonary arterial smooth muscle cell proliferation. With liquid and gas chromatography-based mass spectrometry, a metabolomic study of human microvascular pulmonary arterial vascular smooth muscle cells was conducted in idiopathic PAH (Group 1) individuals before and following therapy with the selective adenosine triphosphate-competitive mTOR inhibitor PP242. So, alterations in fatty acid synthesis, drops in the levels of sugars, amino sugars, and nucleotide sugars intermediates of protein and lipid glycosylation, as well as reduced expression of crucial compounds belonging to the glutathione and nicotinamide adenine dinucleotide (NAD) metabolism were detected. Again, mTOR blockage settled or reversed most of these PAH-specific flaws [60].

Idiopathic PAH is a serious condition at any age of life [61]. Using liquid chromatography–mass spectrometry, higher levels of plasma spermine were found in individuals with idiopathic PAH in comparison with healthy subjects. Spermine was responsible for the increase in the number and migration of pulmonary arterial smooth muscle cells and for the exacerbated unfavorable pulmonary arterial remodeling in rodent models of PAH [62].

Patients with idiopathic PAH are often affected by G6PD deficiency [63]. A metabolomic study of G6PD-deficient male mice detected the activation of metabolic pathways alternative to the pentose phosphate pathway, thus markedly increasing oxidative stress (for example, with the raised activation of myo-inositol oxidase), and the fatty acid pathway, and the reduction of pyruvate production [64].

In a sample consisting of subjects with idiopathic PAH and PAH due to congenital heart disease (both belonging to PAH Group 1), a metabolomic analysis was able to detect a dysregulation in the different metabolic pathways, including lipid, glucose, amino acid, and phospholipid metabolisms pathways, compared to healthy subjects. Among the affected metabolites, a number of molecules from lipid metabolism and fatty acid oxidation (lysophosphatidylcholine, phosphatidylcholine, perillic acid, palmitoleic acid, N-acetylcholine-d-sphingomyelin, oleic acid, palmitic acid and 2-octenoylcarnitine metabolites) proved to be closely linked to PAH [65]. Subjects with an unrepaired ventricular septal defect with or without PAH were analyzed as well. Metabolomics highlighted a hundred and ninety-one differential metabolites between subjects with PAH and those without PAH. As such, raised levels of serotonin, taurine, creatine, sarcosine, and 2-oxobutanoate, and decreased levels of vanillylmandelic acid, 3,4-dihydroxymandelate, 15-keto-prostaglandin F_2α,_ fructose 6-phosphate, l-glutamine, dehydroascorbate, hydroxypyruvate, threonine, l-cystine, and 1-aminocyclopropane-1-carboxylate were noted. These molecules might be used as biomarkers of the disease [66]. The purine, glycerophospholipid, galactose, and pyrimidine metabolism was also markedly altered in PAH induced by congenital heart disease with left-to-right shunt. The levels of these molecules were correlated with the microbes found in the lung microbiota [67]. ^1^H-NMR metabolomics showed that in adult subjects with congenital heart disease and associated PAH, the concentrations of some metabolites like alanine, glucose, glycine, threonine, and lactate were altered. The levels of the last two molecules were strongly correlated with mean pulmonary arterial pressure, pulmonary vascular resistance, and N-terminal pro-B-type natriuretic peptide concentration [68]. Other potential biomarker sable to characterize individuals with PAH induced by congenital heart disease are serum S-adenosyl methionine and guanine [69]. Plasma metabolomics was performed in the peri-operative time of defect repair in subjects with PAH linked to congenital heart disease. Seventeen metabolites responding to shunt correction were found. They might be used as non-invasive markers to predict the outcome of surgery [70].

Even though both idiopathic PAH and PAH induced by connective tissue diseases belong to Group 1 of the disease classification, different metabolic pathways allow the distinction of the two pathological conditions. In fact, patients with a connective tissue disease like systemic sclerosis and PAH had raised concentrations of fatty acid metabolites (including lignoceric acid and nervonic acid), eicosanoids/oxylipins, and sex hormone metabolites [71]. In fact, the main metabolomic feature of connective tissue disease-associated PAH is a dysregulated lipid metabolism (with reduced concentrations of sex steroid hormones and raised concentrations of free fatty acids), which implies a shifted metabolic substrate utilization and the downregulation of mitochondrial beta oxidation [72].

Drug-induced PAH belongs to PAH Group 1. Monocrotaline is one of the drugs able to induce PAH. It is a pyrrolizidine alkaloid that can be found in plants of the Crotalaria genus [73]. Monocrotaline was injected in rats to cause the development of PAH, and nuclear magnetic resonance was utilized to study plasma molecules. Significant changes in the metabolomic profile were detected amid the first 28 days from injection. Twenty distinct molecules were detected. They were mainly entailed in lipid metabolism, glycolysis, energy metabolism, ketogenesis, and methionine metabolism [74]. In another monocrotaline-induced PAH rat model, metabolites extracted from the right ventricular myocardium were detected by using liquid chromatography associated with mass spectrometry. In summary, it was found that disrupted iron homeostasis, glutathione metabolism, and lipid peroxidation related to ferroptosis may cause right ventricular dysfunction induced by monocrotaline. These processes represent potential therapeutic targets in slowing down the right ventricular function decline [75]. Some of these metabolic changes (upregulated glycolysis, increased levels of proliferative markers, disruption in carnitine homeostasis, raised levels of inflammatory and fibrosis markers, and a decrease in glutathione biosynthesis) preceded the development of PAH [76]. A disruption in the urea cycle was found as well, with five metabolites being involved (adenosine monophosphate, urea, 4-hydroxy-proline, ornithine, N-acetylornithine) [77]. The participation of urea cycle metabolism is observed in idiopathic PAH as well [78]. Non-targeted liquid chromatography–mass spectrometry-based metabolomics revealed a key role for nuclear receptor binding SET domain 2 in the onset of PAH linked to the trehalose metabolism and autophagy by means of increasing the di-methylation level of H3K36 [79]. An increased glycolytic dependence and glutaminolysis induction, as well as a reduced fatty acid metabolism, are hallmarks of the disease in monocrotaline rats [80]. In the presence of a dysfunctional right ventricle caused by monocrotaline-induced PAH, the levels of some metabolites (aspartate and glutathione) were upregulated, whereas those of other molecules (phosphate, α-ketoglutarate, inositol, glutamine, 5-oxoproline, hexose phosphate, creatine, pantothenic acid, and acetylcarnitine) were downregulated. This implies alterations in the processes of glycolysis, fatty acid metabolism, oxidative phosphorylation, tricarboxylic acid cycle [81]. Taken together, all the above stated alterations may contribute to the onset of monocrotaline-induced PAH. Atorvastatin seemed to exert a positive action in monocrotaline-induced PAH by reducing the levels of glycogen synthase kinase-3β and sterol regulatory element-binding protein 1 and increasing the levels of hexokinase 2 and carnitine palmitoyltransferase I [82]. In a rat model of PAH, monocrotaline-induced PAH caused changes in the gut microbiome as well, with dysregulation of many molecules of the gut microflora. However, the calcium-sensing receptor antagonist NPS2143 was able to reverse this intestinal flora disorder and alleviate lung injury at least in part [83].

As mentioned above, PAH associated with connective tissue diseases belongs to Group 1 of the disease classification. In this subset, PAH triggered by systemic sclerosis has a poor prognosis. Eighteen patients were studied. Their blood was sampled during right heart catheterization to perform a metabolomic analysis. In those systemic sclerosis patients who developed PAH, a rise in acetate, alanine, lactate, and lipoprotein levels and a decrease in gamma-aminobutyrate, arginine, betaine, choline, creatine, creatinine, glucose, glutamate, glutamine, glycine, histidine, phenylalanine, and tyrosine levels was noted. this implies an impairment in the release of molecules exerting a protective effect on the endothelium [84].

The blockage of TGF β-activated kinase 1 leads to antiproliferative effects, and its combination with the vasodilator agent riociguat increased the drug’s positive action on pulmonary vascular and right ventricular remodeling. Enhancement of taurine, amino acids, glycolysis, and tricarboxylic acid cycle metabolism through the glycine–serine–threonine metabolism was detected by metabolomics [85].

The important metabolic changes detected among the different PAH phenotypes suggest that they may benefit from different therapeutic approaches [86].

The metabolic features of the distinct subtypes of PAH are summarized in Table 2.

#### 3.2.3. New Pathogenetic Mechanisms

By means of a combination of liquid and gas chromatography-based mass spectrometry, the levels of bile acid metabolites, which are compounds derived from the liver and gallbladder, were found significantly raised in PAH lungs, thus suggesting that even bile acid synthesis may be implicated in the development of the disease [87].

Individuals with idiopathic PAH often have reduced oral glucose tolerance. This is because PAH promotes lipid and ketone metabolism at the expense of glucose control. In this respect, plasma metabolomics identified many metabolites: the concentrations of 213 of them were raised, and those of 145 were reduced [88]. In this respect, a blood metabolomic analysis was performed before and after metformin treatment. It showed that lipid metabolites (including dihydroxybutyrate, acetylputrescine, hydroxystearate, and glucuronate) were among the molecules whose levels were the most affected by metformin. An improvement in RV contractility was detected as well [89].

#### 3.2.4. Right Ventricular Function

The right ventricular structure and function are the most important factors determining symptoms and outcomes in PAH patients [90]. The main cause of death among subjects with PAH is right heart failure [91]. However, the right ventricular biology is not well understood. Metabolomics may contribute to reveal it. PAH leads to right ventricular dysfunction, which is characterized by lipid accumulation. With metabolomics, a significant rise in the levels of long-chain fatty acids was found in mutant mice with PAH in comparison with controls, which correlated with the cardiac index. Fatty acid oxidation was impaired as well [92]. In PAH patients, right ventricular failure is linked to metabolic disarrangements in the cardiomyocytes, which are caused by reduced adenylate energy charge, as testified by metabolomics [93].

In the pulmonary arteries of rats affected by PAH, the biosynthesis of phenylalanine, tyrosine, and tryptophan and the linoleic acid metabolism were highly dysregulated, whilst in the right ventricle, arginine biosynthesis and linoleic acid metabolism were significantly dysregulated [94].

In the hypertrophied right ventricle secondary to PAH, a metabolomic research showed that the tricarboxylic acid cycle was less active, owing to the downregulated expression of fumaric acid and malic acid, which, in turn, may be due to the overexpression of adenylosuccinic acid and argininosuccinic acid. These findings imply that an adversely modified branched-chain amino acids metabolism together with a reduced fatty acid oxidation may lead to the downregulation of the tricarboxylic acid cycle [95].

In PAH subjects, trans-right ventricle (with blood sampling from the superior vena cava and the pulmonary artery) metabolic changes exist and are correlated with hemodynamic parameters of clinical outcome such as tricuspid annular plane systolic excursion and pulmonary vascular resistance. These changes involve especially lipid metabolism/lipotoxicity, with long-chain fatty acids accumulation (dicarboxylic acids and acylcarnitines). Such an accumulation implies impaired beta-oxidation in the hypertensive right ventricle and may represent a novel therapeutic target [96].

Assessing the right ventricular function by imaging and without invasive testing is difficult. A study tried to characterize the metabolomic markers of the right ventricular function at rest and during exertion in PAH subjects. Twenty-three patients underwent rest and exercise right heart catheterization. Mass spectrometry-based targeted metabolomics was performed. The levels of thirteen metabolites changed on exertion. Tryptophan metabolism and, in particular, the kynurenine pathway proved to be linked to the right ventricular function. Arginine bioavailability was also crucial in the response to exercise. In fact, a higher arginine bioavailability at rest envisaged more favorable hemodynamic changes on exercise. The subjects with a more aggressive PAH reported a larger increase in arginine bioavailability as a result of exercise compared to those with a less aggressive disease. Again, the metabolite profiles identified by metabolomics were better than the N-terminal B-type natriuretic peptide levels in predicting right ventricular function at rest and on exertion. In summary, some selected metabolites may act as PAH-specific biomarkers, offer an insight into PAH pathophysiology, and be potential targets for specific therapies [90]. A similar study with chromatography–mass spectrometry was performed on pulmonary arterial blood at rest and peak exercise. Compared to the controls, the PAH subjects showed changes involving glycolysis, tricarboxylic acid cycle, fatty acid and complex lipid oxidation, collagen deposition and fibrosis, and nucleotide metabolism [97]. Right ventricular function and exercise performance in PAH were tested in vivo. Tryptophan metabolism, mostly the kynurenine pathway, was found to be linked to right ventricular function. this highlights the pivotal role of arginine bioavailability in the response of the cardio-pulmonary system to exercise. A higher arginine bioavailability at rest predicted a better response to exertion. Patients with a more severe form of PAH showed a larger increase in arginine availability with exercise compared to those affected by less aggressive forms of the disease [90].

While metabolomics showed significant metabolic changes and metabolic reprogramming induced by PAH in dysfunctional right ventricles, metabolic substrate delivery was substantially unchanged. This may imply a possibility for right ventricular recovery [98].

Targeted metabolomics showed that intermittent fasting reduced the right ventricular levels of microbiome metabolites such as bile acids, aromatic amino acid metabolites, and gamma-glutamylated amino acids. So, intermittent fasting exerts a direct action in improving right ventricular function and changing the gut microbiome. As such, intermittent fasting may represent a non-pharmacological way to ameliorate the impaired right ventricular function, a deadly aftermath of PAH [99].

## 4. Discussion

Many metabolomics studies have been released in regard to PH/PAH so far. Regarding the occurrence of PH, which is the first step of the disease, the metabolic cornerstones seem to be a shifting from oxidative phosphorylation to glycolysis and an increase in glutamine uptake. The latter is the most common non-essential amino acid. Glycolysis is a metabolic pathway and an anaerobic producer of energy. It uses two molecules of ATP to produce four molecules of ATP in a kind of investment and pay-off process [100]. However, this shifting process is not enough to meet the high request of energy of the pulmonary vascular cells, which is necessary to sustain their excessive proliferation. So, the tricarboxylic acid cycle is activated as well. During the cycle, intermediate molecules need to be steadily replaced through some complementary pathways. The decomposition of glutamine by the enzyme glutaminase is one of these pathways. It provides intermediate products for the tricarboxylic acid cycle, thus favoring cellular proliferation and inhibiting cellular apoptosis [101]. In some specific forms of PH, other metabolic disarrangements have been detected. For example, some amino acids are involved (such as arginine, which is reduced and is entailed in nitric oxide synthesis, a potent vasodilator whose production is reduced in PH [102]; tryptophan, which is increased and induces pulmonary arterial smooth muscle cell contraction, probably through serotonin synthesis [103]; proline, an anti-fibrotic agent, which is also reduced) [104]. Again, also enhanced oxidative stress is present in PH. It exerts a pivotal role in endothelial dysfunction by increasing the synthesis and release of vasoconstrictor factors like endothelin-1 and reducing those of vasodilator factors such as nitric oxide. Once established, oxidative stress generates reactive oxygen species and leads to the secretion of mitogenic and fibrogenic factors inducing cell proliferation and fibrosis in the vascular wall, resulting in adverse vascular remodeling [105]. Other metabolic abnormalities specifically linked to some forms of PH are the increase in lipolysis (which causes triacylglycerols to break down into their constitutive biomolecules, i.e., glycerol and free fatty acids; the latter are utilized to produce energy), fatty acid oxidation (a process that breaks down fatty acids for energy production), and ketogenesis (another alternative metabolic pathway to produce ketone bodies and then energy from them) [106,107].

All these processes are even exacerbated when PH evolves into irreversible PAH. The hallmarks of the disease progression remain an increase in glycolysis (which is testified by the raised contents of lactate and alanine, the endpoints of glycolytic oxidation) and impaired mitochondrial respiration. Oxidation is exacerbated as well. In this respect, the enzyme xanthine oxidase proved to be overexpressed. This enzyme is needed to generate uric acid through the rupture of purine nucleotides. Uric acid itself, along with the reactive oxygen species produced amid the enzymatic reaction, can exert a harmful action on the pulmonary vasculature [108]. Specific metabolic fingerprints for many of the specific PAH subtypes have been identified. Metabolomics revealed specific metabolic targets for new therapeutic agents as well. The 17 metabolites responding to shunt correction in congenital heart disease patients represent an example of how metabolomics could influence decision making and/or patient monitoring [70].

Epidemiology suggests the influence of gender on the onset of PAH, with a female-to-male ratio of approx. 4:1, depending on the underlying illness. However, notwithstanding the gender imbalance, also males are affected by PAH. Metabolomics applied to the study of PH/PAH is still in its pioneering phase. As such, so far no studies have specifically focused on shedding light about the hormonal influence on the PAH distribution between genders.

Because it is the right ventricular function which influences the disease prognosis, the integration of right ventricular imaging by echocardiography or cardiac magnetic resonance with metabolomic analysis may represent a further advantage when evaluating these patients.

## 5. Conclusions

In conclusion, PH is a condition resulting from a number of different pathological etiologies. Depending on the etiology, therapy can be quite different, but early diagnosis and accurate classification of the disease are crucial. With the recently proposed new definition of PH and related drop in the pulmonary arterial pressure threshold, even more diagnostic problems will have to be faced. Recent genome-wide analyses have shed light on the genetic development of PH/PAH. However, this is not enough. A novel approach to PH/PAH patients’ phenotyping is needed. In this respect, metabolomics may reveal the processes occurring in the body of a patient with PH/PAH. Novel metabolic pathways involved in the pathology, which are opening the way to a personalized medicine approach, have been identified. As such, each specific metabolic fingerprints may constitute a potential target for a specific (and new) therapeutic strategy aimed at amending the metabolic error underlying that particular form of PH/PAH. This explains why metabolomics is a step forward towards personalized medicine or tailored medicine. Although it is not easy, such a search for metabolic pathways leading from the discovery of “biomarkers” to the establishment of “metabolic fingerprinting” has already begun in the field of PH/PAH [109,110].

## Figures and Tables

**Table 1 ijms-24-13227-t001:** Pulmonary hypertension groups with their metabolomic fingerprints.

Group	Metabolomic Fingerprint
Group 1	Reduced concentrations of omega-3 polyunsaturated fatty acids and increased concentrations of omega-6 metabolites *(persistent pulmonary hypertension of the newborn)* Changes in arginine, pyrimidine, purine, and tryptophan metabolic pathways (*pulmonary hypertension induced by drugs*) Disturbances involving fatty acid metabolism, reactive oxygen species, and the extracellular matrix (*pulmonary hypertension induced by pulmonary vein stenosis)*
Group 2	Overexpression of long-chain acylcarnitines, acetylcarnitine, and monounsaturated fatty acids (*pulmonary hypertension induced by heart failure*)
Group 3	Alterations of the levels of lactate, glutamate, and adenosine 5′-triphosphate (*pulmonary hypertension induced by congenital diaphragmatic hernia*) Unbalanced lactate and pyruvate levels (*pulmonary hypertension induced by chronic obstructive pulmonary disease*) Increased malondialdehyde concentrations, reduced glutathione peroxidase and superoxide dismutase activities, changes in purine, arginine, and proline metabolic pathways (*pulmonary hypertension induced by chronic exposure to high altitude*)
Group 4	Glycolysis-derived metabolites, ketone bodies, tricarboxilic acid cycle intermediates (energy imbalance); sphingolipids and lysophospholipids (cellular signaling); rise in fatty acids, glycerol, acyl carnitines, beta-hydroxybutyrate, amino sugars and modified amino acids, and nucleosides levels, decrease in acylcholines and lysophospholipids levels (increase in lipolysis, fatty acid oxidation, and ketogenesis) (*pulmonary hypertension induced by chronic thromboembolism*)
Group 5	No studies have been carried out on this group

**Table 2 ijms-24-13227-t002:** Pulmonary arterial hypertension groups with their metabolomic fingerprints.

Group	Metabolomic Fingerprint
Group 1	Alterations in fatty acid synthesis, drops in the levels of sugars, amino sugars, and nucleotide sugars intermediates of protein and lipid glycosylation, and reduced expression of crucial compounds belonging to glutathione and nicotinamide adenine dinucleotide (NAD) metabolism were detected (*idiopathic pulmonary arterial hypertension*) Changes in spermine metabolism, raised activation of myo-inositol oxidase and the fatty acid pathway, reduced pyruvate production (*idiopathic pulmonary arterial hypertension*) Disruption in the urea cycle (*idiopathic pulmonary arterial hypertension*) Changes in the levels of lysophosphatidylcholine, phosphatidylcholine, perillic acid, palmitoleic acid, N-acetylcholine-d-sphingomyelin, oleic acid, palmitic acid, and 2-octenoylcarnitine metabolites (*idiopathic pulmonary arterial hypertension and pulmonary arterial hypertension induced by congenital heart disease*) Raised levels of serotonin, taurine, creatine, sarcosine, and 2-oxobutanoate and decreased levels of vanillylmandelic acid, 3,4-dihydroxymandelate, 15-keto-prostaglandin F2α, fructose 6-phosphate, l-glutamine, dehydroascorbate, hydroxypyruvate, threonine, l-cystine, and 1-aminocyclopropane-1-carboxylate; changes in purine, glycerophospholipid, galactose, and pyrimidine metabolism; alterations in alanine, glucose, glycine, threonine, and lactate levels; alterations in S-adenosyl methionine and guanine metabolism (*pulmonary arterial hypertension induced by congenital heart disease*) Raised concentrations of fatty acid metabolites (including lignoceric acid and nervonic acid), eicosanoids/oxylipins, and sex hormone metabolites (*pulmonary arterial hypertension induced by connective tissue disease*). Disarrangements in lipidic metabolism, glycolysis, energy metabolism, ketogenesis, and methionine metabolism; disrupted iron homeostasis, glutathione metabolism, and lipid peroxidation related to ferroptosis; disruption in the urea cycle (adenosine monophosphate, urea, 4-hydroxy-proline, ornithine, N-acetylornithine); increased di-methylation level of H3K36; increased glycolytic dependence and glutaminolysis induction, reduced fatty acid metabolism; aspartate and glutathione upregulation, phosphate, α-ketoglutarate, inositol, glutamine, 5-oxoproline, hexose phosphate, creatine, pantothenic acid and acetylcarnitine down regulation (alterations in the processes of glycolysis, fatty acid metabolism, oxidative phosphorylation, and tricarboxylic acid cycle); increased levels of glycogen synthase kinase-3β and sterol regulatory element-binding protein 1 and reduced levels of hexokinase 2 and carnitine palmitoyltransferase I (*pulmonary arterial hypertension induced by drugs*) Rise in acetate, alanine, lactate, and lipoprotein levels and drop in gamma-aminobutyrate, arginine, betaine, choline, creatine, creatinine, glucose, glutamate, glutamine, glycine, histidine, phenylalanine, and tyrosine levels (*pulmonary arterial hypertension induced by connective tissue diseases*)
Group 2 to 5	There are no metabolomic studies on these groups

## Data Availability

Not applicable.

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
