# Peer review of "Metabolomics in Pulmonary Hypertension—A Useful Tool to Provide Insights into the Dark Side of a Tricky Pathology"

_ijms, 2023, doi:10.3390/ijms241713227_

Round 1

Reviewer 1 Report

I clarified what I marked with pink in the original text in this document.

In the first paragraph of the introduction, when it mentions how pathology is identified after 'smooth muscle cells proliferation,' it says hypertrophy. I'm asking if it's cardiac or pulmonary.

The paragraph on page 5: Chronic exposure to high altitude can cause PH development. As such, 1H NMR metabolomics technology was used for serum analysis in an animal model of the disease. 18ß-glycyrrhetinic acid significantly reduced pulmonary arterial pressure and malondialdehyde levels. It increased the glutathione peroxidase and superoxide dismutase activities, thus exerting an antioxidant and anti-inflammatory effect and restoring several metabolic pathways (such as energy, amino acid, and lipid metabolisms) [30]. I suggest it be integrated into the paragraph where other information on high altitude is mentioned. This is the one paragraph after the next.

On page 9: the highlighted text can be integrated in only one sentence. Fuse them.

On page 11: Crotalaria should not be in italics. Crotalaria.

On page 14: the highlighted text can be integrated in only one sentence. Fuse them. And the parenthesis is open, but there is no closure.

Author Response

Dear Colleague,

Thanks for the time spent in reviewing our paper. Unfortunately, the Journal HAS NOT PROVIDED me with the text marked by you with pink. I would be obliged if you could provide me with the pages and lines with a view of amending the text, please. About the raised points:

  • Introduction, first paragraph: “smooth cells proliferation” has been replaced with “hypertrophy”. It refers to “pulmonary” hypertrophy;
  • Page 5: PH related to chronic exposure to high altitude belongs to group 3 (group 5 is a typo). As such, the sentences linked with references [30] and [34, now 31] have been put together;
  • Page 9, regarding the highlighted text to merge into one sentence: as mentioned, I have not got the text marked by you with pink. In the second round of the review would you mind providing me with page and lines, please?;
  • Page 11, “Crotalaria”: the Italics has been replaced;
  • Page 14, regarding the highlighted text to merge into one sentence: again, I have not got the text marked by you. In the second round of the review would you mind providing me with page and lines, please?;

The changes in the text have been highlighted in red. Other changes have been made according to the other Reviewers’ suggestions.

Reviewer 2 Report

The manuscript you submitted for review is an excellent work of background collection where metabolomics is essential for developing pulmonary hypertension research. The approach used is appropriate, which makes the background provided by the authors very valid. The methodology used is adequate in the systematization of the bibliographic collection, which allowed conclusions very relevant to the development of the manuscript. However, the authors should add the following ideas so that these conclusions are adequate for developing the manuscript's ideas.

1.- The authors should better develop the difference between PH and PAH. Since the development of the manuscript is based on the characterization and exposition of the same ideas in both pathologies (or what, according to the authors) are early and late stages of the same pathology (idea that emerges from the reading of the manuscript). With the development of this differentiation, the reader will have better tools to follow the reading of the manuscript.

2.- The sections on pathogenesis, new pathogenic mechanisms, and right ventricular function are very well developed. However, both sections on specific subtypes should be better developed because the references used compare subtypes of pulmonary hypertension with healthy individuals (references 22 and 24, for example) do not make differentiation or a study to establish differences between subtypes of pulmonary hypertension. This point is crucial to know whether the variables analyzed serve to differentiate the different subtypes. This subsection must be considerably improved to give coherence to the subtitle.  

In the rest of the manuscript, the authors adopt interesting approaches that provide an innovative look at metabolomics in pulmonary hypertension research.

Author Response

Dear Colleague,

Thanks for the time spent in reviewing our paper. Your very useful suggestions have been taken care adequately, namely:

  • Introduction: a few notes about the difference between PH and PAH have been added;
  • For the sake of clarity, at the end of the sections about PH and PAH pathogenesis, two Tables summarizing the related findings have been added.

The changes in the text have been highlighted in red. Other changes have been made according to the other Reviewers’ suggestions.

Reviewer 3 Report

remarks to the authors:

After an intriguing and promising title, suggesting insight into the dark side of a tricky pathology. This was quite a tough read for the non inborn error of metabolism expert, and non systems-biology expert. Unfortunately I  fall into  neither of these 2 catagories.

This review seems to be a quite comprehensive summary of what has been researched with respect to metabolomics in relation to PH/PAH, which may be a valuable refererence for researchers in this field. However I feel this text  would greatly benefit from illustrative tables  and/or figures and  clinical examples. In the abstract the authors claim specific metabolic fingerprints allow to characterise some of the specific PH and PAH subtypes. It would make this review much more accessible if they provided one or two examples where the use of metabolomics in a clinical setting would lead to  a more precise diagnosis, that might influence clinical management. They state ’Metabolomics is a step forward (on the way to?) personalised medicine’ I reckon that is generally speaking correct, but I believe it is a missed chance that the authors fail to put forward an example of how metabolomics could personalise the approach to PH/PAH in an individual patient. What would you advocate?

Major problems with this review: the only figure in the supplementals the prisma flow diagram I did not understand: The arrows are wrong or the numbers are  incorrect. It simply  does not add up. See page 3/23 data extraction  and prisma flow chart:

 You start with 544 potentially eligible studies and end up with 460 studies after removal of 84 duplicates, so far the flow diagram corresponds to the text. Then you remove 313 studies after screening of the title and abstract; for reasons beyond my comprehension you end up with 231 studies for full-text assessment ( instead of 147). Then you remove 128 ( according to the text for not meeting the inclusion criteria and a furher 18 for not being available full text from online libraries. Thus you, deus ex machina end up with 85 eligible studies, which you expand to 96 after having checked the references of the first 85 included studies. Possibly this is just a simple misunderstanding; and me doing bad calculations. But it is evident that the text and flow chart are not reconcilable. In the flow chart you remove another 267 studies for not meeting inclusion criteria, which would have left you with -36 studies for inclusion. Maybe it seems that I am too much fixated on numbers, but this touches on the core of doing unbiased reviews.

Page  4/23 3.1.2. specific PH subtypes:

You mention classification of PH in different groups 1 to 5 . It would be extremely helpful if you provided a sort of table  with the different groups, what they consist of and their specific metabolomics signatures. That would make this paper much more accessible.

Page 6/23 3.1.3  new pathogenetic mechanisms

I am aware of the association of PH with thyroid disorders. But I do not understand the connection :

Cardiovascular diseases are often seen in patients with PH. In this respect…?

Maybe start the alinea with: Serum levels of…

As hyperthyroidism is not a cardiovascular condition per se.

Page 6/23 3.1.3  new pathogenetic mechanisms

2nd alinea: Untargeted metabolomics was carried out using liquid chromatography and mass spectrometry on what exactly?

Page 10/23  first heading:

Patients with idiopathic PH are often affected by G6PD deficiency ( please provide a proper reference). The association is plausible given the assumption that hemolysis might contribute to PAH causation, but somewhat counterintuitive given the fact that there is a female predomimance in PAH and G6PD is an X-linked recessive disorder in man.

As an extension to this question I would like to pose the following question: In relation to the female to male bias in PAH, hormonal  factors have been put forward to explain this female excess, I was rather hoping your metabolomics  review would shed a light on this mysterious sex bias. But actually you report very little on hormonal alterations and how they might influence disease causation/progression.

Page 10/23   end of 2nd heading: you mention that 17 metabolic markers respond to shunt correction, would this maybe be an example where metabolomics in the clinic could influence decision making and/or patient monitoring?

Page 10/23  last  heading: I do not understand you compare iPAH with connective tissue disease induced PAH ( both group 1): you state that patients with systemic sclerosis-PAH have a different metabolic profile as compared to connective tissue  disease associated PAH.  I may be wrong but I figured systemic sclerosis falls within the group of MCTD ( mixed  connective tissue diseases! Please clarify?

Page 11/23 last 2 lines You state that Monocrotaline-induced PAH causes changes in the gut microbiome as wel, with dysregulation of many molecules of the gut flora. You probably mean to say that injection of monocrotaline cause changes.. and not monocrotaline induced PAH causes changes…. And  you add that the compound NPS2143 could reverse these intestinal changes. Could you explain in the text why this influence on the intestinal flora is relevant for your review on the metabolomics of PH/PAH?

Author Response

Dear Colleague,

Thanks for the time spent in reviewing our paper. We understand that the paper may be difficult to read for those who are not expert in metabolism errors. However, the manuscript has been submitted to a Special Issue entitled “The Impact of Altered Metabolism on Cardiac Development and Disease”. Your very useful suggestions have been taken care adequately, namely:

  • The specific metabolic fingerprints for each form of PH and PAH have been summarised in Tables 1 and 2 now. Each specific metabolic fingerprints may constitute the potential target for a specific (and new) therapeutic strategy aimed at amending the acquired (“not-inborn”) metabolic error underlying that particular form of PH/PAH. This explains why metabolomics is a step toward personalized medicine or tailored medicine. It has been added in the Conclusion paragraph at the end of the paper;
  • Thanks for detecting the mistake in the flow chart. It was simply related to “copy and past” from the flow chart of a previous manuscript. Now the text and the flow chart match each other;
  • Page  4/23, 3.1.2. specific PH subtypes: a Table has been added, according to your suggestion (Table 1). The same was done at the end of the paragraph on “specific PAH subtypes” (Table 2).
  • Page 6/23, 3.1.3 new pathogenetic mechanisms: “Cardiovascular diseases” is a typo. It has been replaced with “Other kind of diseases”;
  • Page 6/23, 3.1.3 new pathogenetic mechanisms, second paragraph: untargeted metabolomics was used in a diabetic-like condition in an in vitro setting made of pulmonary artery smooth muscle cells amid hyperglycaemia. It has been added;
  • Page 10/23 first heading, “Patients with idiopathic PAH are often affected by G6PD deficiency”: a proper reference has been added. You are right, G6PD deficiency is an X-linked recessive disorder, thereby males usually manifest the abnormality and females are carriers. Conversely, epidemiology suggests the influence of gender on the onset of PAH with a female to male ratio of approx. 4:1, depending on the underlying disease pathology. However, notwithstanding the gender imbalance, of course also males can be affected by PAH. The study was carried out on male mice.
  • Metabolomics applied to the study of PH/PAH is still in its pioneering phasis. As such, so far there no studies specifically focused on shedding light on the hormonal influence on the different PAH distribution between genders. It has been added in the Discussion section;
  • Page 10/23, end of 2nd heading: thanks for your advice. We have added the suggested example in the Discussion section;
  • Page 10/23, last heading: the study compared idiopathic PAH with PAH due to systemic sclerosis, which is a connective tissue disease. Idiopathic PAH and PAH due to systemic sclerosis both belong to Group 1 PAH, but metabolomic fingerprints are different;
  • Page 11/23, last 2 lines: we meant that in a rat model of PAH, monocrotaline-induced PAH causes changes in the gut
    microbiome as well with dysregulation of many molecules of the gut microflora. However the calcium-sensing receptor antagonist NPS2143 was able to at least in part reverse this intestinal flora disorder and alleviate lung injury. The sentence in the text has been slightly modified for the sake of clarity.

The changes in the text have been highlighted in red. Other changes have been made according to the other Reviewers’ suggestions.